# High-Sensitivity Detection of IgG Operating near the Dispersion Turning Point in Tapered Two-Mode Fibers

**DOI:** 10.3390/mi11030270

**Published:** 2020-03-05

**Authors:** Bing Sun, Yiping Wang

**Affiliations:** 1Guangdong and Hong Kong Joint Research Centre for Optical Fibre Sensors, College of Physics and Optoelectronic Engineering, Shenzhen University, Shenzhen 518060, China; graduate_sunbing@163.com; 2Key Laboratory of Optoelectronic Devices and Systems of Ministry of Education and Guangdong Province, Shenzhen University, Shenzhen 518060, China

**Keywords:** biosensor, optical fiber sensor, two-mode fiber, sensitivity

## Abstract

The conventional methods for monitoring IgG levels suffer from some apparent problems such as long assay time, multistep processing, and high overall cost. An effective and suitable optical platform for label-free biosensing was investigated by the implementation of antibody/antigen immunoassays. The ultrasensitive detection of IgG levels could be achieved by exploiting the dispersion turning point (DTP) existing in the tapered two-mode fibers (TTMFs) because the sensitivity will reach ±∞ on either side of the DTP. Tracking the resonant wavelength shift, it was found that the fabricated TTMF device exhibited limits of detection (LOD) down to concentrations of 10 fg/mL of IgG in PBS solution. Such immunosensors based on DTP have great significance on trace detection of IgG due to simple detection scheme, quick response time, and miniaturization.

## 1. Introduction

Immunoglobulin G (IgG) is a short chain of amino acids connected by peptide bonds, and each molecule has two antigen bindings, which it uses to protect us from invading organisms. IgG, which represents approximately 75% of serum antibodies in humans, is the most common type of antibody found in blood circulation. Clinically, the levels of IgG can generally considered be indicative of an individual’s immune status to particular pathogens. Conventional methods include enzyme linked immunosorbent assay (ELISA), which is the gold standard for monitoring the IgG levels because of its sensitivity and accuracy. However, it suffers from some apparent problems such as long assay time, multistep processing, high overall cost, and requirement of sophisticated instrumentation. Meanwhile, the development of in-situ and real-time detection devices is an innovative field in applied research and healthcare diagnostics. Especially label-free biosensors based on optical fibers with high sensitivity for the recognition of low concentrations of analytes have become a research hotspot [1]. These optical fiber biosensors can be based on many constructions such as fiber gratings [2,3,4], special fibers [5,6,7], novel fibers with mechanical treatment [8,9], etc. Furthermore, the advancement of nanotechnology plays an essential role in the field of biosensing, benefiting from advanced materials and nanostructures such as transducer elements or reporters [10]. Liu et al. explored a graphene oxide nanosheets functionalized dual-peak long period grating-based biosensor for immunosensing detection, and the limit of detection (LOD) is 0.05 mg/mL [11]. It is undeniable that the relative sensitivity and LOD are excellent due to the high RI sensitivities.

On the other hand, one kind of microfiber working at the turning point can show infinite refractive index (RI) sensitivities. Luo et al. reported a high RI sensitivity of 10,777.8 nm/RIU, but the microfiber was concentrated on the utilization of single-mode fiber configuration [12]. Similarly, researchers have exploited optical microfiber coupler sensors working near the turning point of the effective group index difference between super modes to achieve high RI sensitivity [13,14,15,16]. For example, Zhou et al. demonstrated an ultrasensitive label-free optical microfiber coupler biosensor with diameter of 1.0 μm for detection of cardiac troponin I based on interference turning point effect. Experimental experience revealed that the fabrication and integration of such an optical microfiber coupler in a chamber is challenging [15]. In this study, a high-sensitivity biosensor based on a tapered two-mode fiber (TTMF) sandwiched between two single-mode fibers (SMFs) was experimentally demonstrated. Experimental results indicate that the RI sensitivity was also significantly enhanced when it works around the dispersion turning point. Furthermore, this biosensor exhibited a sensitive response to IgG levels over the concentration range of 5–500 fg/mL in phosphate-buffered saline (PBS) buffer. Besides, a detection limit of 10 fg/mL in PBS buffer was obtained, which is useful for practical applications in clinical diagnostics. Moreover, the structure is easy to fabricate, highly stable, and easy to integrate with a plastic chamber.

## 2. Theoretical Analysis

A ring–core two-mode fiber (TMF, http://www.yofc.com) was employed. Its cross-section and index profile can support fundamental (*HE*_11_) and second-order (*HE*_21_) modes. The spatial profile of such modes is depicted in Figure 1. Tapering the two-mode fiber is recognized as a straight and efficient tool to achieve a uniform modal interference mainly involving the *HE*_11_ and *HE*_21_ modes. Thus, we propose a tapered two-mode fiber (TTMF)-based biosensor structure, which consists of an input SMF, a down-taper region, a TTMF section, an up-taper region, and an output SMF. Note that the number of modes existing in the TTMF is determined by its diameter (D_w_) and the external surrounding (n_ext_) [17]; thus, the section of TTMF needs to be sophisticatedly designed to develop a desired interferometer.

Owing to the cosine function of phase difference between the *HE*_11_ and *HE*_21_ modes, the evolution of the transmission spectra can be illustrated according to the two-mode interferometric theory as follows:(1)$$I=I_{1}+I_{2}+2\sqrt{I_{1}I_{2}}\cos\left(\Phi \right)$$
where $\Phi =2\pi {\Delta n}_{\mathrm{eff}}L/\lambda$ is the phase difference between the *HE*_11_ and *HE*_21_ modes, ${\Delta n}_{\mathrm{eff}}$ is the effective RI difference of the *HE*_11_ and *HE*_21_ modes, *L* is the effective length of the taper, and $I_{1}$ and $I_{2}$ are the intensities of the *HE*_11_ and *HE*_21_ modes, respectively. Thus, the interference dip ($\lambda_{m}$) can be defined.
(2)$$\lambda_{m}=\Delta n_{\mathrm{eff}}L/\left(m+1/2\right),m\mathrm{is}\;a\;\mathrm{positive}\;\mathrm{integer}$$

Furthermore, the response of the transmission interference dip *λ_m_* to external RI ($n_{\mathrm{ext}}$) can be deduced through several mathematical treatments of Φ and its relationship (*S*) can be expressed as
(3)$$S=\frac{d\lambda_{m}}{dn_{\mathrm{ext}}}=\frac{1}{\Gamma }\frac{\lambda_{m}}{∆n_{\mathrm{eff}}}\frac{d∆n_{\mathrm{eff}}}{dn_{\mathrm{ext}}}$$
where the determinative parameters of *S* are the wavelength λ*_m_*, the RI-induced variation of index difference $d\Delta n_{\mathrm{eff}}/dn_{\mathrm{ext}}$, and the dispersion factor Γ ($1-\lambda_{m}/∆n_{\mathrm{eff}}\times d∆n_{\mathrm{eff}}/d\lambda_{m}$). Furthermore, Γ and $d\Delta n_{\mathrm{eff}}/dn_{\mathrm{ext}}$ are dominated by *d_w_* and *n_ext_*. We previously reported that the values of the dispersion factor Γ increase as the microfiber becomes thinner [18]. It is interesting to note that the dispersion turning point (DTP) appears when the dispersion factor Γ approaches zero. According to Equation (3), at DTP, Γ approaches zero and the RI sensitivity is enhanced significantly to be ±∞.

## 3. Results

### 3.1. Refractive Index Sensing

Initially, we utilized the flame-brushing technique to fabricate the TTMF structure, where three stages assembled with electric motors, a hydrogen generator, and an oxyhydrogen torch were employed. The taper sample can be fabricated by sophisticatedly defining the parameters pulling speed and hydrogen flow rate, which can be found in the previous report [19]. Thus, we carried out experiments to investigate the spectral characteristics and the RI sensing capabilities of the TTMF. As illustrated in the top of Figure 2, a taper with a width of ∼4.7 μm was produced by the flame-brushing technique while the length of the uniform waist and tapered transition regions were ∼7 and ∼11 mm, respectively. The tapered TMF obviously helped to establish a stable Φ, leading to the interference status. Note that the DTP did not disappear in the given wavelength range when it was surrounded by air. By contrast, the corresponding spectrum showed evident changes when the structure was immersed in an index-matched liquid, with the DTP appearing around the wavelength of 1570 nm, as shown in the bottom of Figure 2. We believe that the dispersion factor plays a dominate role in this phenomenon.

The abovementioned ultrasensitive characteristics have been proposed previously [12,13,14,15,16,17], where minor refractive index variation could induce evident shift of the dip around the DTP. However, inducing a minor refractive index change is challenging. In addition, it causes uncertainty and measurement errors when various glycerol solutions with different weight concentrations were prepared [19]. Thus, in this experiment, we induced minor refractive index changes with different views. Initially, an index-matching liquid, which possessed a RI of 1.34 at 589.3 nm and a thermal coefficient of d*n*_fluid_/d_T_ = −3.38 × 10^−4^/°C, was employed, and then the TTMF was immersed into such liquid packaged in a flow cell. Next, we precisely controlled the surrounding temperature by placing the flow cell into a high-precision column oven (LCO 102) with an accuracy of 0.1 °C. In addition, a differential thermocouple (UNI-T UT320) was used to monitor the surrounding temperature. We did not need to account for the temperature cross-sensitivity of the TTMF as the thermal expansion of fused silica is of the order of 10^−7^/°C. Accordingly, we established a temperature control system to obtain the minor refractive index change during the experiment. Figure 3a illustrates the transmission spectra shift with the ambient refractive index. The experimental results reveal that the resonant dip on the left side of the DTP shifted linearly toward a shorter wavelength, i.e. the so-called “blue” shift, while the resonant dip on the right side of the DTP showed “red” shift with increase of the ambient refractive index. It is well known that the increase of $n_{\mathrm{ext}}$ can normally generate a larger index increment for the *HE*_21_ mode than that for the *HE*_11_ mode, producing $d\Delta n_{\mathrm{eff}}/dn_{\mathrm{ext}}\;$< 0. On the other hand, we had Γ > 0 since the group velocity of the *HE*_11_ mode is larger than the *HE*_21_ mode on the left side of the DTP. Thereby, a negative *S* was enabled, corresponding to a “blue” shift (Dip A) of wavelength with an increase of $n_{\mathrm{ext}}$ while Dip B showed the opposite trend. Furthermore, Figure 3b presents sensitivities of the resonant dips closest to the DTP (Dips A and B), which provided the highest sensitivities of −48,772 and +79,283.7 nm/RIU, respectively. More significantly, the resonant dips on both sides of the DTP shifted in the opposite directions with the variations of the ambient refractive index. Thereby, an improved sensitivity of ∼128,055.7 nm/RIU was achieved, which is a competitive result compared to the state-of-the-art fiber-optic refractive index sensors.

### 3.2. Detection of IgG Levels

#### 3.2.1. Functionalization of Biosensor

To further demonstrate the applicability and to improve upon the TTMF-based biosensor, we used it for the quantitative detection of IgG in PBS. This biodetection process is depicted in Figure 4. Prior to the functionalization, we thoroughly cleaned the microfiber surface to remove any traces of glycerin, first with distilled water and then with methanol. Immediately after, the microfiber was cleaned with distilled water and immersed into 0.1 M KOH solution for 10 min to create hydroxyl(-OH) groups on the surface. The surfaces were then washed with distilled water. Next, the fiber biosensor surface was prepared by incubation for 30 min with 1 wt% poly (diallyldimethylammonium chloride) (PDDA, Mw: 200000–350000, 20 wt% in H_2_O, purchased from Sigma-Aldrich, St. Louis, MO, USA). After washing with distilled water, the microfiber was immersed into 1 wt% polyacrylic acid solution (PAA, Mw: ~100000, 35 wt% in H_2_O, purchased from Sigma-Aldrich) to get final modification with individually spotted capture proteins. The surfaces were also washed with distilled water. The immobilization of biomolecules on device surface is an important step in biosensor development. The covalent immobilization of anti-IgG on fiber surface might lead to improper orientation by masking antigen-binding sites. This shortcoming can be circumvented by using heterobifunctional cross-linkers of EDC/NHS combination. The microfiber was immersed into a mixture (1:1) of EDC (1-ethyl-3- (dimethylaminopropyl) carbodiimide hydrochloride) and NHS (N-Hydroxysuccinimide) in 0.01 M PBS buffer for 30 min. Then, the sensor was washed thoroughly with PBS. Subsequently, functionalized fiber was incubated with anti-IgG at a concentration of 50 mg/L for 1 h to immobilize the antigen onto the fiber surface. Blocking to reduce subsequent non-specific binding was then performed by incubating the sensor with a blocking solution (3% bovine serum albumin (BSA) in PBS, pH 7.4) for 30 min. The biosensor chip was then washed with PBS for 5 min.

#### 3.2.2. Experimental Setup

IgG and anti-IgG binding was performed by injecting and incubating 50 μL of IgG solutions at different concentrations and then rinsing the biosensor using PBS solution. The PBS rinse stage was used to establish the baseline associated to each increasing IgG concentration and to remove the unbound antigens from the previous solution. The experimental setup for the immunosensing performance characterization is illustrated in Figure 5. A broadband source (ASE) with a low polarization spectral range of 1250–1650 nm (www.optical-source.com) was used as the light source and its transmission spectra were monitored by an optical spectrum analyzer (OSA) with a minimum resolution of 0.02 nm in that spectral range. Moreover, the biosensor was embedded in a flow cell with the facilities for inserting and removing biological samples, and both ends of the flow cell were sealed with UltraViolet (UV) curable epoxy.

As illustrated in Figure 6a, we could easily notice that the DTP appeared at around 1400 nm when evaluating its immunosensing performance on the right side as a wavelength shift in real time. Note that the interferometric spectra share the general envelope with that immersed into the index-matched liquid while inducing several ripples across the interference. We noticed that the polyelectrolyte layers immobilized onto the surface of the microfiber could lead to exciting higher-order modes, thus probably introducing multiple interferences into the biosensor. Nevertheless, we could track the DTP region, further locating the resonant dip on the right side of the DTP, although it was not easy to determine at first. However, the evident contrast of the resonant dip happened as the concentration was increased. Besides, some experience tracking the resonant dip was necessary. Furthermore, the spectra gradually showed “blue” shift when the IgG concentration was increased. Then, the anti-IgG were gradually exhausted and the spectral shift reached a plateau at the IgG concentration of 500 fg/mL due to the saturation effect, which can be considered a complete antibody–antigen procedure process. As a result, the shifts of the dip were fitted by a rotational function with a high coefficient R^2^, as shown in Figure 6b.

#### 3.2.3. Evaluation of Antibody-Antigen Binding Interaction

To further show the dynamic process, the real-time response to IgG with concentrations ranging from 5 to 500 fg/mLwas obtained by continuously recording the interferometric dip near the DTP, as shown in Figure 7. We noticed that the spectra hardly showed any shift for an IgG concentration of 5 fg/mL, as we believe such concentration is beyond the limit of detection of the proposed biosensor. However, the procedure was intense and robust for an IgG concentration of 10 fg/mL, and a maximum wavelength shift of ∼1 nm was recorded during this stage. Accordingly, we conclude that a limit of detection down to the concentration of 10 fg/mL of IgG in PBS solution can be achieved and the concentration range is no more than 500 fg/mL.

## 4. Discussion

A dynamic procedure for the detection of a concentration of 10 fg/mL of IgG can be understood. Initially, the amounts of anti-IgG were rich and then it matched that of IgG, as illustrated in the enlarged window of Figure 7. Thus, the procedure reached a plateau due to saturation effect. On the other hand, the enlarged view in Figure 7 fully indicates a complete antibody–antigen conjugation process. Obviously, the processes tended to be stable after antibody–antigen binding for several minutes, which was recorded by the dynamic processes. We could easily achieve the resonant dip, thus concluding that the antibody–antigen procedure required approximately 15 min. Thus, the stability of the proposed biosensor could be revealed. Furthermore, the reusability can probably be attained by short injections of the regeneration buffer after each immune response, as the immune process is consuming and irreversible. Similarly, Zhou et al. demonstrated that the output light intensity can return to the initial level after each regeneration cycle [15]. We will further clarify this in future work.

The reproducibility mainly refers to the TTMF fabrication and its successive functionalization. Initially, we could skillfully master the geometric parameters of the microfibers, such as the diameter and length of the transition region, by the moving speeds of the flame and stages to guarantee the reproducibility at this stage. Admittedly, its functionalization, which involved multiple and tedious procedures, is a challenge. Furthermore, polyelectrolyte layers immobilized onto the surface of the microfiber make it fragile. We understand this situation and aim to improve it in follow-up experiments.

## 5. Conclusions

We comprehensively investigated the temperature response of the TTMF. Utilizing the DTP, both ultrahigh refractive index sensitivities (with a negative sensitivity of −48,772 nm/RIU and a positive sensitivity of +79,283.7 nm/RIU) were experimentally demonstrated. Furthermore, relying on the excellent index sensing capability, an ultra-sensitive biosensor was also demonstrated. The proposed TTMF device exhibited limits of detection down to concentrations of 10 fg/mL of IgG in PBS solution. Moreover, it is expected that this novel microfiber can serve as a necessary step towards potential applications for fast and accurate quantification of biomarker concentrations in clinical settings.

## Figures and Tables

**Figure 1 micromachines-11-00270-f001:**
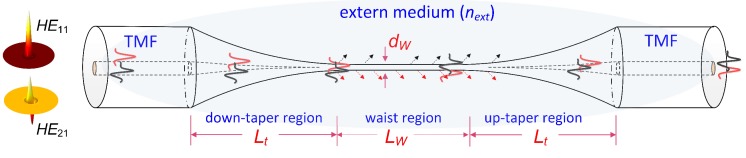
Schematic diagram of the sensor. Quasi-adiabatic transition tapers provide continuous mode evolution from *HE*_11_ and *HE*_21_ modes when the two modes launched into the fiber are collected at the fiber output, respectively.

**Figure 2 micromachines-11-00270-f002:**
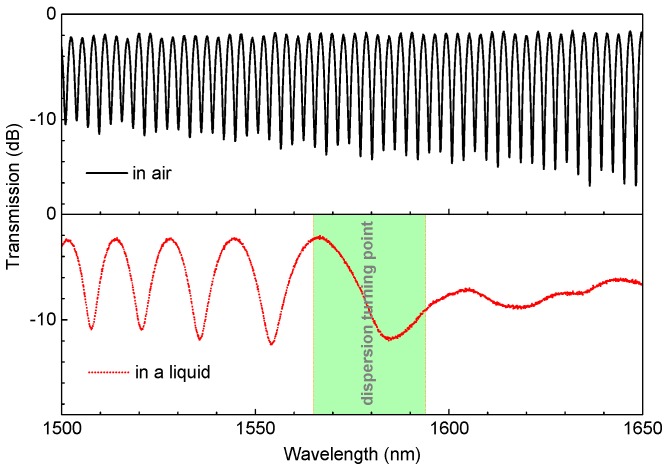
Variation of the interference spectra of the TTMF when immersed in air and a liquid.

**Figure 3 micromachines-11-00270-f003:**
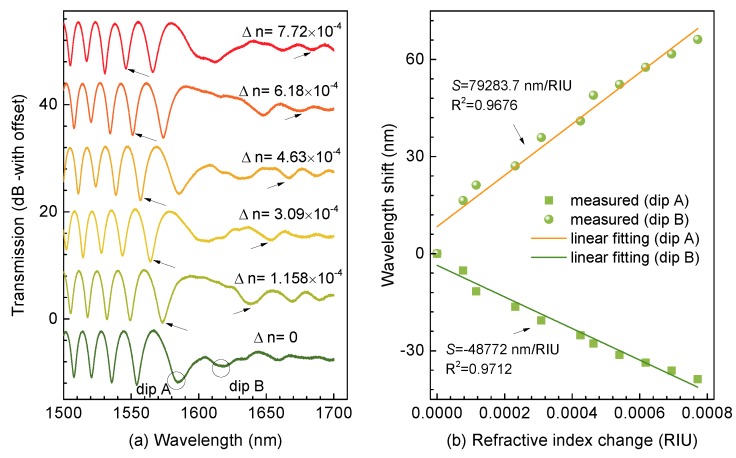
(**a**) Variation of interferential spectra as the ambient refractive index. (**b**) Wavelength shifts of Dips A and B versus ambient refractive index.

**Figure 4 micromachines-11-00270-f004:**
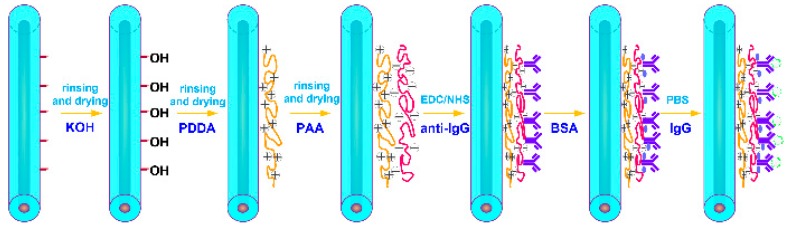
PDDA and PAA modification procedure for IgG immunoassay.

**Figure 5 micromachines-11-00270-f005:**
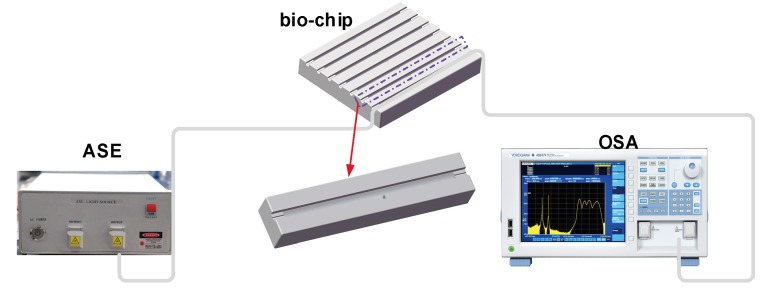
Experimental setup for the biosensing structural characterization.

**Figure 6 micromachines-11-00270-f006:**
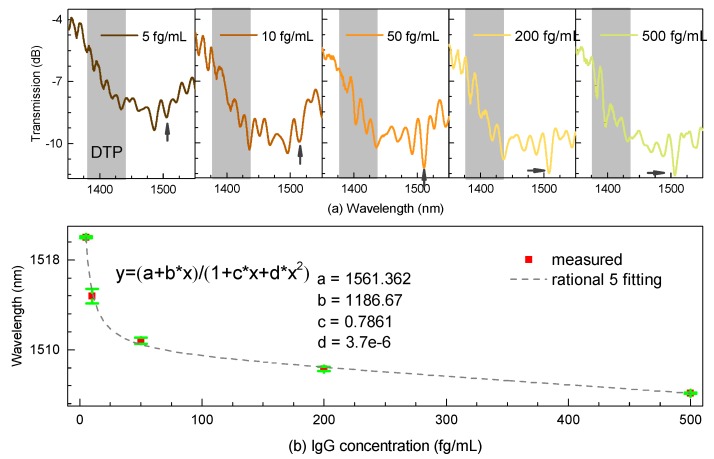
(**a**) Transmission spectral responses near the DTP. (**b**) Resonant shift with different concentrations of IgG.

**Figure 7 micromachines-11-00270-f007:**
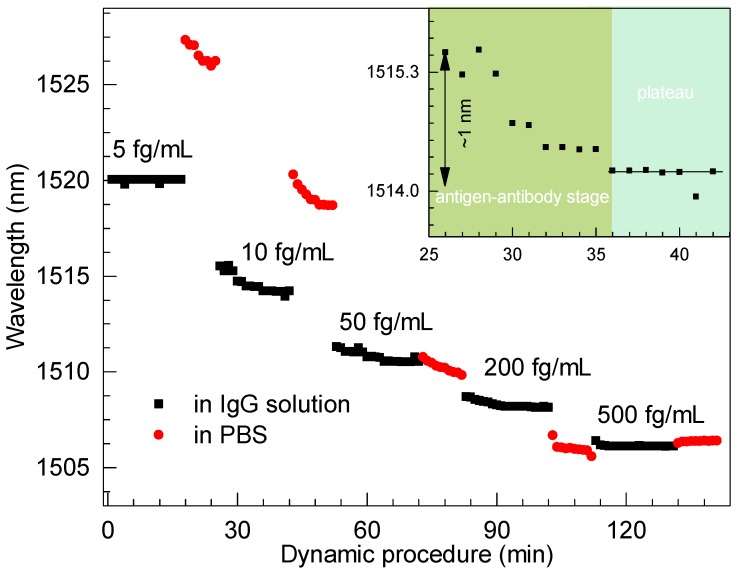
Spectral shift of the biosensor at different stages of IgG binding events.

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
