# Peer review of "High-Sensitivity Detection of IgG Operating near the Dispersion Turning Point in Tapered Two-Mode Fibers"

_micromachines, 2020, doi:10.3390/mi11030270_

Round 1

Reviewer 1 Report

It would be good to define what IgG is at early point in the manuscript. What is the refractive index change associate to 10fg/ml what is the correspondent wavelength shift? Fig.3 (a) is it theoretical or experimental. It is not clear how the change in the refractive index was induced? The authors mentioned using only one refractive index matched liquid with an RI of 1.3400 at 589.3 nm. Has this liquid been diluted? What is the refractive index at 1550nm? There are no information given about manufacturing the TTMF sensor, parameters about the length and the diameter of the sensitive region is not given. Line 98 it is still not clear how the change in the refractive index was introduced? Is it though an index matching liquid that is also very sensitive to temperature? If so what are the calibration data for that fluid (i.e refractive index Vs temperature). How these values of the sensitivities were obtained? Section 3 needs rewriting and more details about the sensor characterisation needs to be given, additional section describing the fabrication of the sensor should be included prior to section 3. Fig. 6 shows the DTP around 1400nm, however the graph doesn’t really show DTP, it only shows some ripples or interference patterns over 100nm range. While for Fig.3 the width and the shape of the DTP is different.  In my opinion, more work needs to be done and more data and information needs to be added to make this manuscript publishable.

Reviewer 2 Report

The authors present a sensor to detect IgG using a tapered two-mode fiber working in the vicinity of the turning point of the effective group index difference. The paper has some potential however, in its present form, it should not be accepted for publication and the reasons are as follows:

English must be improved; The theory section is not sufficient, namely concerning equation 1. Also lacks the interpretation of the experimental results (movement of the resonance wavelengths) The characteristics of the taper are not given; There is no temperature control during the experiment. Temperature cross-sensitivity was not discussed; Resonances at the right side are very shallow and, therefore, difficult to determine the wavelength. Reproducibility, stability and reusability of the sensors not discussed. Although the OSA resolution may be 0,01 nm, it was not the real resolution for the 200 nm wavelength range.

Although in the broad sense it may be acceptable, this may not be the best journal to publish this kind of research.

Round 2

Reviewer 2 Report

I acknowledge the effort done by the authors to improve their manuscript. The main question were answered and therefore the paper may be published. However, I would like to mention that English still need to be improved and highlight the fact that the reproducibility, stability and reusability of the sensors were not discussed and are important topics in these kind of sensors.